# A Hierarchical Approach for Traffic Sign Recognition Based on Shape Detection and Image Classification

**DOI:** 10.3390/s22134768

**Published:** 2022-06-24

**Authors:** Eric Hsueh-Chan Lu, Michal Gozdzikiewicz, Kuei-Hua Chang, Jing-Mei Ciou

**Affiliations:** Department of Geomatics, National Cheng Kung University, Tainan City 701, Taiwan; p66097068@gs.ncku.edu.tw (M.G.); p68111012@gs.ncku.edu.tw (K.-H.C.); jing-mei@geomatics.ncku.edu.tw (J.-M.C.)

**Keywords:** traffic sign, object detection, image recognition, deep learning, computer vision

## Abstract

In recent years, the development of self-driving cars and their inclusion in our daily life has rapidly transformed from an idea into a reality. One of the main issues that autonomous vehicles must face is the problem of traffic sign detection and recognition. Most works focusing on this problem utilize a two-phase approach. However, a fast-moving car has to quickly detect the sign as seen by humans and recognize the image it contains. In this paper, we chose to utilize two different solutions to solve tasks of detection and classification separately and compare the results of our method with a novel state-of-the-art detector, YOLOv5. Our approach utilizes the Mask R-CNN deep learning model in the first phase, which aims to detect traffic signs based on their shapes. The second phase uses the Xception model for the task of traffic sign classification. The dataset used in this work is a manually collected dataset of 11,074 Taiwanese traffic signs collected using mobile phone cameras and a GoPro camera mounted inside a car. It consists of 23 classes divided into 3 subclasses based on their shape. The conducted experiments utilized both versions of the dataset, class-based and shape-based. The experimental result shows that the precision, recall and mAP can be significantly improved for our proposed approach.

## 1. Introduction

Traffic sign recognition is a topic that has been present in the scientific community for years. The main problem that all researchers face is achieving the best possible accuracy while maintaining the complexity and similarity with the real-life environment of the dataset. Traffic sign detection and classification are pivotal in various engineering solutions that have been gaining in popularity recently. The main two applications of traffic sign recognition are Intelligent Speed Assistance (ISA) [1] and autonomous vehicle camera systems. As the ISA system recognizes a traffic sign, the speed limit data is obtained and then compared with the vehicle speed. The driver is promptly notified about the speed of the vehicle and whether the vehicle is breaking the speed limit. Recent research has determined that in urban areas, the number of car crashes causing casualties is doubled for each 5 km/h over the limit [2]. This one finding provides enough information to realize the impact traffic sign detection and classification can have on people’s safety when used with the ISA system. The second main application of traffic sign recognition is its usage in the development of autonomous vehicles. According to statistics [3], most road accidents take place as a result of a lack of response time to dynamic traffic events that take place on the road. This problem can be addressed with the implementation of self-driving cars and their automated systems aimed to detect these events and react to them in due time. Maneuvering through all the real-time events that can occur on the road requires correctly identifying the traffic signs faced by the automated vehicle, classifying them to ensure that the system understands their contents and then reacting to them in an appropriate manner. In addition, traffic landmarks, such as road markings, traffic signs and light poles, serve an important role in indicating routes, directions, locations, etc., to all drivers and pedestrians. However, with long-term use, these landmarks may be affected by external forces or natural factors [4]. These external factors can lead to damage and, in severe cases, endangerment of the drivers and pedestrians when using the road. Therefore, regular road inspections are extremely important and necessary. However, traditional detection methods are usually accompanied by certain risks in addition to requiring a lot of workforce and time [5]. Therefore, this paper focuses on applying automatic identification signs to road inspection to reduce human resources needed and time costs.

Traffic sign recognition has two varying approaches that have been utilized by researchers in recent years: two-stage networks, that focus mainly on improving the accuracy of detection and classification, such as R-FCN [6], Faster R-CNN [7], Mask R-CNN [8], etc.; and one-stage networks, that focus on improving the speed of the aforementioned tasks, such as YOLO [9], SSD [10], etc. Both approaches have their advantages and disadvantages and perform differently in certain circumstances. One-stage detectors, such as YOLO, are great for detecting small objects and are resilient to changes in image quality caused by weather conditions or object obstruction but struggle when the number of classes to distinguish is too big or they are similar to one another. Two-stage detectors perform well with the task of classification, but their accuracy falls dramatically when the traffic sign is too far away, blurry or partially covered by other objects or as a result of weather conditions. Accuracy of detection and classification translates directly to better interpretation of the road by the systems utilizing traffic sign recognition while focusing on the speed of the process takes into consideration the limited computing capability of hardware present inside autonomous vehicles [11]. For the problem of traffic sign detection, there are various obstacles that must be overcome; the most important one is, as previously stated, the effects of weather conditions obstructing the traffic sign. Fog, rain, snow or even bad lighting conditions can severely decrease the chance of successful detection. That is why it is so pivotal to take these conditions into consideration when developing any object detection solution. The second stage, classification, focuses solely on recognizing the exact image present on the traffic sign. Its complexity is based mostly on the number of classes in the dataset, as well as the level of similarity between them.

In this paper, the focus is mainly placed on trying to achieve the best possible accuracy for two separate tasks: traffic sign detection and traffic sign classification. Before the development of the module, the team collected papers on relevant research topics of the two utilized stages, conducted an evaluation of included methods and unified the two stages of detection and classification as one task of traffic sign recognition. The first stage starts with detecting the position of the target object. First, a one-stage or two-stage object detector is used to detect the mask or bounding box of the target object and to identify its shape. After obtaining one of the features in the first stage, the second stage aims to solve the classification problem by training another CNN model to further identify the content of the detected object, in our case, the traffic sign. Our proposed approach uses the most popular object detectors: Mask R-CNN, in the first stage, while the second stage uses a relatively freshly introduced model called Xception [12]. This approach makes sure that both models are utilized in relation to their main strengths to achieve the best possible accuracy. The main contribution of this work is the proposal of a novel two-stage framework to detect and recognize traffic signs. In the proposed network, we first detect the traffic signs and their corresponding shapes and then divide them into 23 classes based on the information present on the traffic sign. The used dataset was specifically developed by the authors to be used in Taiwan; thus, it consists of traffic signs present in the country to make sure its utilization would work correctly in real-life situations. The experimental results based on real-world datasets show that our proposed framework is superior to the state-of-the-art object detection models Mask R-CNN and YOLOv5 [13] in terms of detection and classification accuracy.

The rest of this paper is organized as follows. The second section reviews some of the important research that was pivotal to writing this work, especially when it comes to choosing the correct approach and comparing its results with other researchers. Section 3 officially defines the problem, how we aim to solve it in this paper and the methodology and structure of the dataset. The fourth section addresses experiment and evaluation, including the training and testing of three chosen models, Mask R-CNN, Xception and YOLOv5. The last section of this work focuses on concluding our research and discussing potential advancements of this solution that could be used in the future.

## 2. Related Work

When it comes to image processing, the implementation of a Convolution Neural Network (CNN) is widely used in image recognition, classification, feature extraction, semantic segmentation and other applications. After a major breakthrough, different CNN model architectures are developed, such as Fast R-CNN, Faster R-CNN, Mask R-CNN, YOLO, etc., whose purpose is not only to identify and classify images but also to mark the object in the image. The relative position in the self-driving system achieves the effect of assistance in the self-driving system, enabling the system to recognize road signs. CNN for image recognition can be used in various fields. In addition to classification, it is more desirable to obtain information such as the relative position and approximate size of the object in the image. Using Sliding Windows is the simplest concept: Fixed large and small boxes are scanned across the entire image and identified one by one using CNN. In order to achieve better results, the usage of different size boxes is required, which slows the process by a noticeable margin. In this section, this paper reviews some methods of using image recognition and detection of traffic signs. This section is divided into three parts, including databases, classification and object detection.

### 2.1. Traffic Image Database

In terms of image databases, since the definitions of signs in Taiwan are different from those in foreign countries, there are many Taiwan-specific signs that are not available in foreign databases, and Taiwan has not yet published a database for users to download. Therefore, the authors collect images of Taiwan signs to build a database. A detailed description is provided in Section 3. This paper refers to three articles to determine the number of collected images we use. Zhanwen Liu et al. [14] used the Chinese traffic sign database Tsinghua-Tencent 100K (TT100K) for model training in 2021, with 9176 training images. The author divided the 221 types of signs in TT100K into 25 groups through the clustering algorithm, and the final accuracy reached 89.7%. Valentyn Sichkar et al. [15] used a two-stage task to identify traffic signs. The first stage used an object detector to detect object location and shape, while the second stage used a classification model to identify the type of signs. The first-stage image database used the German Traffic Sign Detection Benchmark (GTSDB), with 630 training images and four categories, with an accuracy of 97.22%; the second-stage image database used the German Traffic Sign Recognition Benchmark (GTSRB), with 50,000 training images and 29 categories, with an accuracy of 86.8%. Shao-Kuo Tai et al. [16] used the Taiwan traffic sign database for model training in 2020. The database was collected in the Taichung area, with 630 training images and four categories, with an accuracy of 93.3%.

### 2.2. Image Classification

In this section, this paper introduces several well-known convolutional neural networks. VGG, proposed by Karen Simonyan et al. in 2014 [17], is one of the classic networks in the CNN classification problem and won second place in the 2014 LSVRC classification competition. The smaller stack of 3 × 3 kernels enables the model to dig deeper for better accuracy. Its regular design, concise and stackable convolutional blocks, show good performance on other datasets and are widely used by people. A common VGG is VGG16, whose architecture uses five convolutional layers and three fully connected layers. In the same year, GoogLeNet [18] was proposed by the Google team. Christian Szegedy et al. won first place in the ILSVRC classification competition in 2014. In the practice of pursuing the depth of the model, the authors adopted an inception module for the first time. An Inception module is a method of grouping the filters of the convolutional layer. The filters of different scales in the same convolutional layer can achieve better and more useful feature values. Because GoogLeNet is composed of multiple inception modules, it is often referred to as an Inception Network. Until now, Inception has been derived from the Inception V4 version.

After the success of GoogLeNet and its introduction of the inception module, Francois Chollet proposed another model based on it, but instead of utilizing this solution like the authors of GoogLeNet, he modified it, thus creating Xception. Based on the architecture of InceptionV3 and using depthwise separable convolution, he replaced the original Inception module, as shown in Figure 1. The main difference between Xception and its predecessors is that the general network directly uses 3D convolution and regards space and depth as related to each other, while Xception first performs 2D convolution on each channel and then 1D convolution on the result while also ensuring the overall space and depth remain independent. Xception has higher accuracy on ImageNet than other networks (VGG-16, ResNet-152 [19], Inception V3). Its Top-1 accuracy is 0.79, and its Top-5 accuracy is 0.945. In the end, the second stage model of this paper selects Xception. Finally, the model chosen for traffic sign classification is Xception.

### 2.3. Object Detection

One of the two models chosen for solving the traffic sign detection problem is YOLOv5, the newest addition to the widely known YOLO family. Proposed by Joseph Redmon et al. in 2016, compared with the other model utilized in this work, Mask R-CNN, YOLO focuses not necessarily on achieving perfect accuracy but mostly on detection speed. It is an important factor, especially when it comes to autonomous driving TSD implementation. Due to its speed and relatively low computation requirements, it could potentially be used in self-driving cars possessing limited hardware capabilities. The core of object detection in YOLO is to divide the picture into several S × S-shaped tiles, as shown in Figure 2. Each square predicts the confidence score and type of the detected object with itself as the center and finally outputs the square with the highest total score to find the bounding box of the target object. As shown in Figure 3, YOLO designs the network as end-to-end, which not only makes training easier but also manages to make training speed faster.

With the update of the YOLO architecture, four architecture documents of YOLOv1, v2, v3 and v4 have been released, while v5 is still considered an experimental model with no official paper released yet. Compared with the v1 version, YOLOv2 [20] has been improved in three aspects: more accurate prediction, faster speed and the maximum number of objects to be identified has been increased. Joseph Redmon et al. proposed a joint training mechanism in 2017 and used separate detection datasets and classification datasets to simultaneously train Object Detectors, which learn the location of objects while enhancing the number of used categories for further classification. Joseph Redmon and others continued to improve YOLO and published YOLOv3 [21] in 2018. Compared with the previous two generations, YOLOv3 has higher accuracy and introduced ResNet (Residual Network) and FPN (Feature Pyramid Networks) [22] networks to solve the gradient problem encountered when general neural networks are deepened. They also combined feature maps of different scales to improve the effect of small object detection. In 2020, Joseph Redmon, the father of YOLO, announced his withdrawal from the field of computer vision. YOLOv4 [23] was proposed by Alexey Bochkovskiy and others to improve various parts of YOLOv3, maintain the speed, greatly improve the detection accuracy and reduce the hardware requirements. After the release of YOLOv4, another version of YOLO was being worked on by Glenn Jocher of ultralytics. YOLOv5 was released in 2020 and is once again an open-source model. While the architecture is very similar to its predecessor, YOLOv4, there has been no official paper release from the model’s authors.

YOLOv5 is the newest addition to the YOLO family. For the first time, the model framework has been changed from the previously used Darknet-53 to PyTorch. This change enables researchers to easily connect their models to online databases for quick result storage, visualization and analysis. Its architecture is virtually the same as YOLOv4, which has been praised for its simplicity, speed and low computational cost. In this method, we train the model by splitting our dataset into training and validation with a ratio of 0.8/0.2 using all available 23 classes, as well as all possible image sources, both GoPro and taken by mobile devices. All experiments are automatically recorded and visualized using the Weights and Biases website. YOLOv5 has four different versions of the model with varying depths. (Figure 4). After analysis of their performance and time required for the training process, YOLOv5m has been chosen as the model used in experiments as it offers comparable accuracy to the deeper versions but with a much lower training time. This is an important factor as YOLOv5 is used mostly for performance comparison with Mask R-CNN.

Due to the model’s experimental state and very recent introduction, there have not been many scientific works regarding its use, especially when it comes to traffic sign detection. We managed to find two papers using YOLOv5 as the object detector. In the first case, it was used as a means of detecting bridge cracks with the goal of monitoring civil infrastructure. With only one category to detect but also adding a variety of shapes, bridge cracks can allow the author to achieve a mean average precision value of 0.88 [24], which suggests that the model’s performance is highly dependent on the number of classes in the dataset. The one scientific work where YOLOv5 was used for traffic sign detection managed to achieve a mean average precision of 0.72 with the dataset consisting of 10 classes with around 200 images per class in the training dataset [25]. Images used in this work comprise typical traffic sign detection specialized pictures. However, there are no close-up captures, which could potentially be used in the second stage of image classification. The main problem with using YOLOv5 for solving the problem of traffic sign recognition is its inability to perform at the desired level with a high number of classes.

Many studies have proposed CNN networks that can perform instance segmentation. For example, released in 2017, Mask R-CNN can output a mask with each object’s outline. Mask R-CNN is the latest product of the R-CNN series. Previous methods in the R-CNN family could only output bounding boxes. First, the concept of R-CNN [26] is to use the selective search [27] on the image to select multiple proposed regions and use CNN to extract features from the candidate regions’ values, which are then classified by SVM. However, the whole process costs time, so Fast R-CNN appeared. It directly extracted features of the entire image to speed up the time, which is different from extracting each proposed region’s feature by R-CNN. Then, Faster R-CNN proposed the Region Proposal Network to replace selective search, making it faster. Finally, Mask R-CNN was proposed. It is a fully convolutional network that can not only detect objects but also perform instance segmentation; therefore, it is widely used. For example, Tabernik et al. [28] used Mask R-CNN as a basic model and proposed some improvements to detect more kinds of traffic signs. As for other road identification issues, some articles use Mask R-CNN as the basic model for identification. For example, Singh et al. [29] used Mask R-CNN for road damage detection; the authors achieved a mean F1 score of 0.528 with an IoU of 50% in real-world road images acquired with a smartphone camera. Malbog et al. [30] focused on pedestrians and used Mask R-CNN to detect pedestrian crosswalks. Xu et al. [31] presented a simple and effective Mask R-CNN algorithm for more rapid detection of vehicles and pedestrians. The authors adjusted the backbone network and FPN to effectively improve the speed and mAP. While Zhang et al. [32] discussed the topic of vehicular accidents, the author proposed a vehicle–damage–detection segmentation algorithm, which improves the efficiency of solving traffic accident compensation problems through a self-made dedicated dataset and improved Mask R-CNN model.

In summary, there have been various scientific works to adapt and then improve object detection and image recognition. The main conclusion drawn from these papers is that one-stage detectors are fantastic when it comes to detecting small objects on images of varying quality and taken in different conditions. However, they are not good enough to classify the image present on the traffic sign with the desired accuracy. Dividing the process into two stages, detection and classification, has been a common strategy in recent years and is used in our work as well.

## 3. Methodology

In this paper, we propose a hierarchical recognition approach based on object detection and image classification for Taiwan traffic signs. Figure 5 show the framework of our proposed approach. The whole method is divided into two stages. The first stage is a Mask R-CNN model to detect traffic signs according to their shape. The second stage uses the previous stage’s results to obtain the real category of traffic signs through Xception.

### 3.1. Dataset Creation and Annotation

In this paper, we decided to test two different approaches to traffic sign detection and classification. Because of the lack of a sufficiently expanded dataset of Taiwanese traffic signs, we decided to create our own dataset. It consists of 11,074 images of 23 traffic sign classes (Table 1) taken via mobile phone cameras and a GoPro camera mounted inside a car. All mobile photos were taken from 10 different angles and distances to create as realistic an environment as possible. Due to the varying distances resulting in different sizes of traffic signs on the image, the dataset can be used both for detection and classification purposes. The dataset consists of 8 circular traffic signs (6 prohibitory, 2 mandatory), 14 triangular signs and one traffic sign classified as “other”. All images were collected and labelled manually by the project members using an open-source online annotator, VIA (VGG Image Annotator).

In the case of Mask R-CNN, the model needs the outline points of the object to generate the mask. We once again use the “VGG annotator tool” to create an annotation file. Use the “Polygon” method in the tool to select objects and add labels (Figure 6). After saving it as a .json file, it can be imported into the model. Therefore, in addition to the name of the image, the .json file also contains the x-coordinates and y-coordinates of the outline point of each traffic sign in the image.

Next, we crop each single traffic sign image based on the minimum and maximum values of the x and y coordinates in the .json file. Since Xception is a classification model, we only need to store images in different folders according to the labels when cropping (Figure 7).

### 3.2. Our Approach

Our proposed approach is divided into two stages. The first one focuses solely on detecting the shape of the traffic sign. To do that, we utilize a well-known CNN model, Mask R-CNN (Figure 8). It is important to notice that this stage does not recognize all 23 classes included in the dataset but only three distinct shapes that appear in it, triangular, circular and rectangular. The goal of the second stage is to correctly classify previously detected and divided shape traffic signs into the aforementioned 23 classes. According to the output of Mask R-CNN, each traffic sign is extracted from the original image and then used as an input of another CNN model, Xception, which is the final part on our way to classification.

The Xception architecture is a linear stack of depthwise separable convolution layers with residual connections (Figure 9). This makes the architecture very easy to define and modify. It uses a special module called an Inception module. The typical Inception module first looks at cross-channel correlations via a set of 1 × 1 convolutions, mapping the input data into 3 or 4 separate spaces that are smaller than the original input space, and then maps all correlations in these smaller 3D spaces via regular 3 × 3 or 5 × 5 convolutions (Figure 10). With the addition of Xception as the second stage of the experiment, we expect it to perform better than the one-stage detector, YOLOv5. Mask R-CNN should be accurate enough to correctly detect the shapes of the traffic signs, but it will most likely struggle with image classification. Xcpetion model should be able to work on the extracted features and correctly classify the image present on the previously detected traffic signs.

## 4. Experimental Evaluation

In this section, the paper will describe the data format required by the model and the training parameters and then compare the performance and robustness of individual models with our approach. Then, we describe our approach in detail and finally, we experiment with the impact of training data size and image resolution on the model.

### 4.1. Experimental Settings

Regarding the dataset, after collecting the imagery necessary for the creation of the dataset and labeling it using the VGG annotator tool, the team needed to prepare the data for training. All used models utilize different annotations formats, as well as use different image resolutions. In the case of YOLOv5, all annotations had to be converted to the so-called “YOLO format”, which is a format universally used by all YOLO models. The main principle of this format is the design of bounding boxes and how they are portrayed in the annotation files. Bounding box coordinates must be normalized in “xywh format”. If boxes are in pixels, we can obtain coordinates by dividing x_center and width by image width and y_center and height by image height. In the YOLO format, an annotation file is created for every single image in the dataset, where each row relates to a single detected object and is translated to the class, x_center, y_center, width and height of the detected object. To perform the tasks of annotations preparation, an online open-source annotation converting tool, Roboflow, is used (Figure 11).

The training parameters used for all YOLOv5 experiments are the same unless stated otherwise in the experiment description. Image resolution: 416 × 416 (YOLO requires the resolution to be dividable by 32). Batch size: 32 (bigger batch size resulted in training failure due to insufficient GPU memory). Epoch number: 1000. Learning rate: 0.01 (base YOLOv5 learning rate). Weight decay: 0.0005. Hardware: GeForce RTX3090, Intel Core i9-10900K CPU 3.70GHz, 64GB RAM. For Mask R-CNN, image resolution: 1024 × 1024. Batch size: 1. Epoch number: 500 (set 100 in our approach). Learning rate: 0.001 (base Mask R-CNN learning rate). Weight decay: 0.0001. Trainable layers: all. Steps per epoch: 100. Hardware: RTX2080 Ti, Intel Core i9-9900K CPU 3.50GHz, 64GB RAM. Additionally, for Xception, image resolution: 224 × 224. Batch size: 64. Epoch number: 10. Learning rate: 0.001. Trainable layers: all. Optimizer: Adam. Loss function: categorical_crossentropy. Use the same hardware as Mask R-CNN.

Below are the results of these experiments with explanations of each one of them and what goals the researchers tried to achieve with them. In all experimental results, we use “our approach” to represent our proposed model. At the same time, we also test using only Mask R-CNN and YOLOv5 to detect and classify traffic signs, which are represented by “Mask R-CNN” and “YOLOv5”.

### 4.2. Performance Comparison

The first experiment aimed to detect and recognize all 23 used classes using both previously introduced methods in order to compare the given performance. The obvious expectation for this experiment was to see that only one detector, YOLOv5 and Mask R-CNN, are not good enough to correctly detect traffic signs and classify their content. A large number of classes, the difference in image sources, their type and the overall size of the used dataset create significant difficulty for any available model. From the results shown in Table 2, the accuracy of both YOLOv5 and Mask R-CNN models is not satisfactory for traffic sign applications. Xception, however, performs well, considering that the dataset consists not only of capturing from close distance images where the content of the traffic sign is clear but also small objects captured from a large distance with a lower-resolution GoPro camera. The above results are very much what the team was expecting to see, with the size and scope of the dataset unsurprisingly proving to be a massive challenge for using only one detector.

To make sure the results are legitimate, we conducted a five-fold cross-validation experiment and calculate the standard deviation of the results for each model. From Table 3, Table 4 and Table 5, the results of each fold are similar, so we did not consider cross-validation in other experiments.

### 4.3. Model Robustness

Next, our dataset was obtained from two different devices: GoPro and mobile phone. We tried to verify that this makes the model more robust. Therefore, we set up the experiment according to the following different situations: (1) training and testing images from the same device; (2) training and testing images from different devices. From the results shown in Table 6, because the number of training images was reduced, lower accuracy may be obtained. However, since the test images are also from the same device, the accuracy was only slightly lower. Additionally, from the results shown in Table 7, when the training and testing images come from different devices, the accuracy of Mask R-CNN and YOLO is significantly lower. Although the accuracy of our approach dropped slightly, it still maintained high accuracy.

### 4.4. Details of Our Approach

The chosen way to achieve the best possible accuracy is to combine two methods and utilize their strengths while eliminating weaknesses by forwarding the task where the model underperforms to the second one, which can complete them with much better accuracy. In the next experiment, Mask R-CNN was used only to detect the traffic signs and their shapes, thus leaving the task of classifying the content of the traffic signs into 23 categories to Xception. As shown in Table 8, the accuracy was expected to massively top the one achieved in the first experiment of this paper as Xception no longer had to detect small objects on GoPro imagery. As expected, the performance of our two models improved drastically when the whole process was divided into two stages, with each model focusing solely on one of them. This approach allowed us to focus on each model’s strength to the fullest.

### 4.5. Impact of Training Data Size

In the next experiment, we wanted to see how the number of training images would affect the performance of our models. We performed five experiments on each model, each time adding 2000 images. As shown in Table 9, the accuracy of both YOLO and our approach increased with more training images. However, for Mask R-CNN, more images still could not help it improve the accuracy. Additionally, our approach had good performance even when the training images were very few.

### 4.6. Impact of Image Resolution

The last experiment we conducted focused on the effect of changes in image resolution on model performance. The resolutions used in this experiment were multiplications of the base resolution commonly used by the model. We multiplied the base resolution by 0.5 and then by 1.5. In the case of YOLOv5, the base resolution was 416 × 416 pixels, and for Mask R-CNN, it was 1024 × 1024 pixels. Additionally, from the results shown in Table 10 and Table 11, the larger the image, the clearer and more visible the traffic signs should be, thus, making it easier for the model to detect the traffic sign and correctly classify its content.

## 5. Conclusions and Future Work

Traffic accidents are an unfortunate part of our everyday life. The threat they pose to our health and safety, not to mention the chaos they cause, is something we should be actively trying to reduce to an absolute minimum. The development of ISA and self-driving vehicles creates an opportunity for these problems to be solved in the foreseeable future but also increases the importance of making these solutions safe, accurate and fully reliable. To improve the accuracy of these systems, we introduced a novel approach utilizing a two-stage solution to the problem of traffic sign detection and classification separately. In this paper, we solved the problem of traffic sign detection using the Mask R-CNN deep learning model and compared its performance with another state-of-the-art work, YOLOv5. Classification proved to be too much of a challenge for the one-stage detector to solve with acceptable accuracy. Due to that, another solution was introduced, focusing only on dividing traffic signs detected by Mask R-CNN into 23 classes. To ensure that our solution would be applicable in real-life problems in Taiwan, a new dataset was created consisting of over 11,000 images captured with various devices, from different distances and in varying atmospheric conditions. The achieved results show the superiority of our method compared to other state-of-the-art models when it comes to the accuracy of predicting the correct traffic sign class. In the end, we were able to achieve a precision of 98.45% for classes that fit into triangular signs and 99.73% for those whose shape was circular. This slight difference in precision could be attributed to the fact that the number of triangular traffic signs in the dataset is higher than that of circular, thus, making it more difficult for the model to classify.

This work provides a solid foundation for future improvements and augmentations. One of the most important aspects of traffic sign detection in relation to autonomous vehicles is its reliance on hardware built into the car. It is obvious that this kind of onboard computer does not possess the same computational capabilities as professional computers suited for training and implementing deep learning models. Our solution could be improved further by putting additional focus on making the model much faster and less reliant on specific hardware. Another thing to consider is the introduction of even more traffic sign classes. Twenty-three is a respectable number that is a challenge to any classification model, but in real-life, this number can be much higher. The contents of the Taiwan traffic signs are divided into geometric graphics, digits and Chinese characters. The difficulty ranges from simple to difficult: pure geometric graphics, pure digits and Chinese characters. Composite signs are the most difficult to deal with, for example, the sign with “where to go” seen on the highway, as they contain both graphics and text. In this paper, we discussed the identification of traffic signs with geometric graphics. In the future, we will try to identify digits, Chinese characters and even more difficult composite signs to continuously expand the number of traffic sign classes. Lastly, some environmental conditions were not present when collecting the imagery for the dataset. Due to that, it is unknown how it would perform, for example, in heavy rainfall or dense fog.

## Figures and Tables

**Figure 1 sensors-22-04768-f001:**
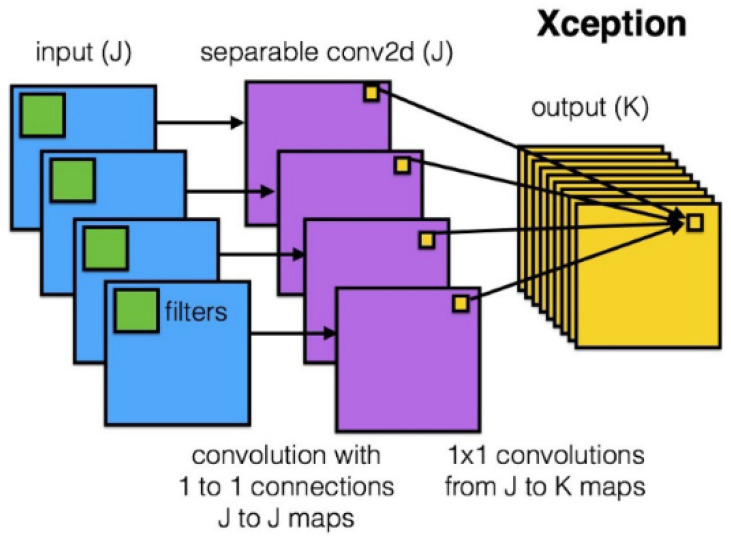
Depthwise separable convolution (Reprinted from ref. [12]).

**Figure 2 sensors-22-04768-f002:**
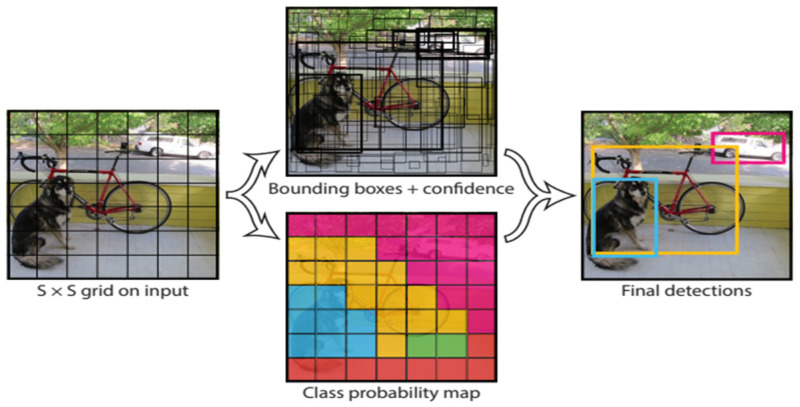
Bounding box generated by YOLO (Reprinted from ref. [9]).

**Figure 3 sensors-22-04768-f003:**
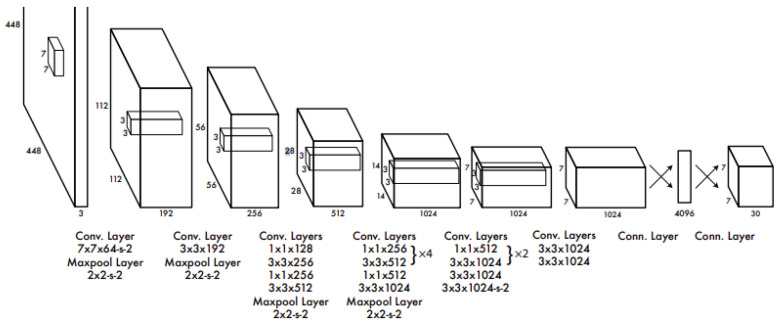
YOLO network architecture diagram (Reprinted from ref. [9]).

**Figure 4 sensors-22-04768-f004:**
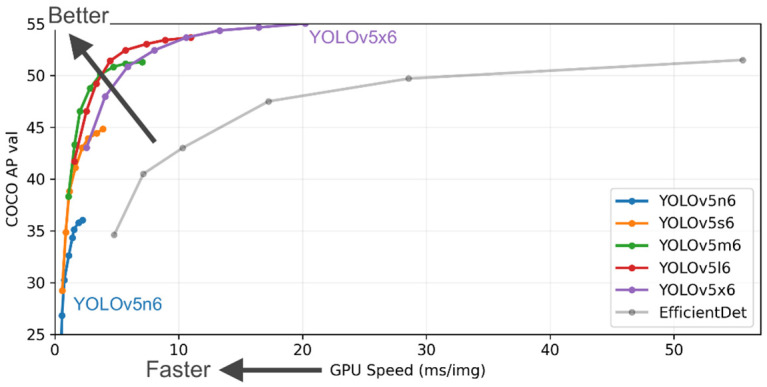
YOLOv5 performance metrics (Reprinted from ref. [13]).

**Figure 5 sensors-22-04768-f005:**
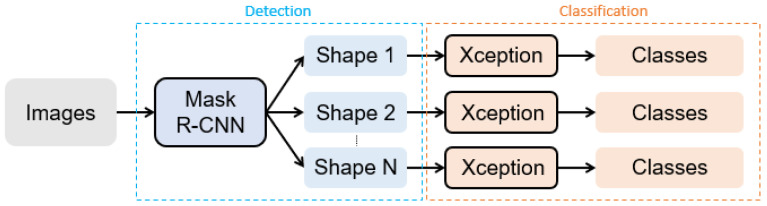
The framework of our proposed approach.

**Figure 6 sensors-22-04768-f006:**
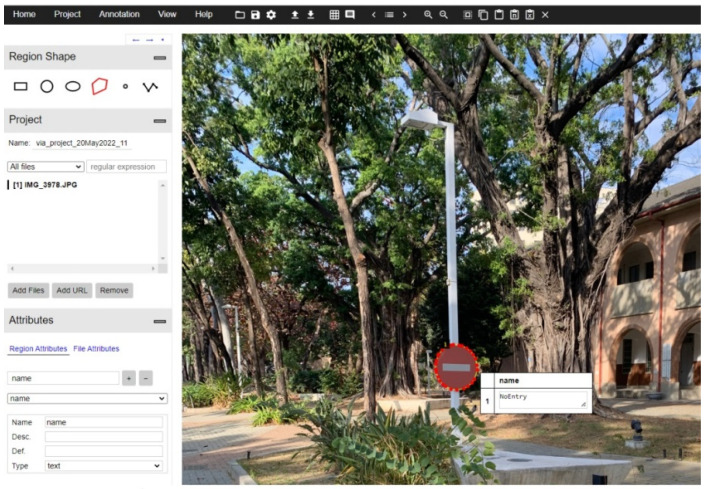
Dataset preparation using VGG annotating software.

**Figure 7 sensors-22-04768-f007:**
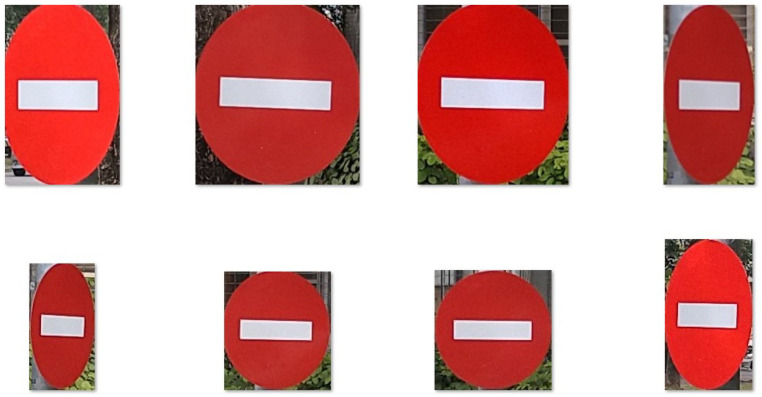
“No entry” traffic sign image cropped for use in Xception.

**Figure 8 sensors-22-04768-f008:**
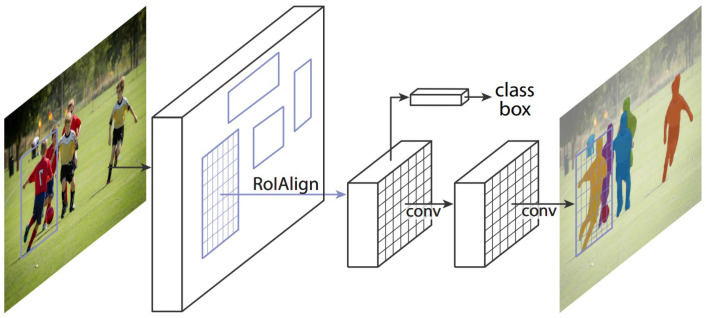
Mask R-CNN framework, for instance, segmentation. (Reprinted from ref. [8]).

**Figure 9 sensors-22-04768-f009:**
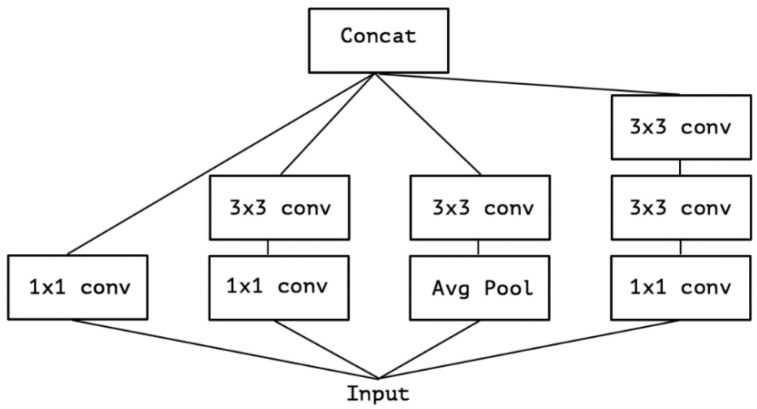
A typical Xception module (Reprinted from ref. [12]).

**Figure 10 sensors-22-04768-f010:**
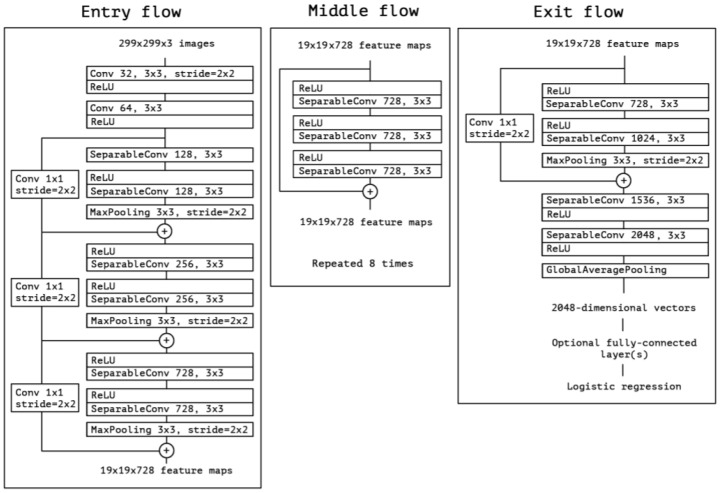
Architecture of the Xception model (Reprinted from ref. [12]).

**Figure 11 sensors-22-04768-f011:**
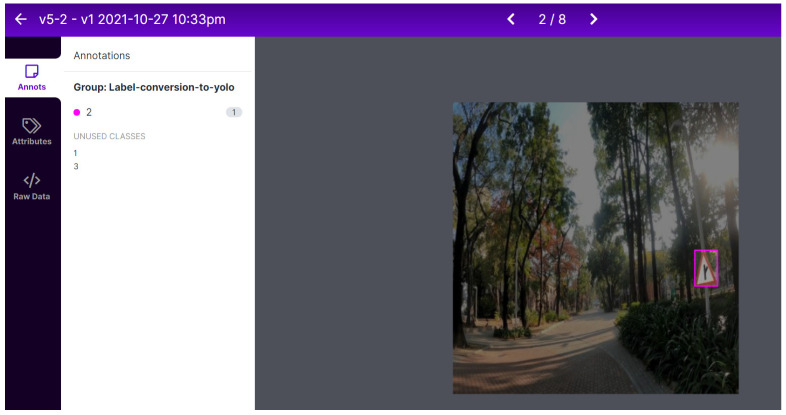
Annotations conversion to YOLO format via Roboflow.

**Table 1 sensors-22-04768-t001:** Presentation of traffic signs in the created dataset (Chinese means metric ton and meter).

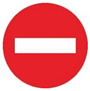 No entry	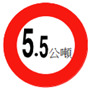 Max vehicle weight	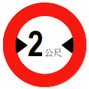 Max vehicle width	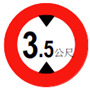 Max vehicle height	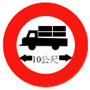 Max vehicle length
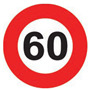 Speed limit	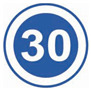 Minimum speed	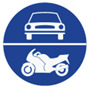 Road-designated vehicles		
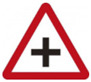 Forked road (1)	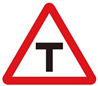 Forked road (2)	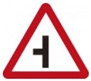 Forked road (3)	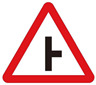 Forked road (4)	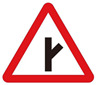 Forked road (5)
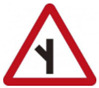 Forked road (6)	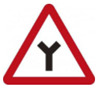 Forked road (7)	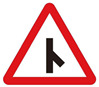 Forked road (8)	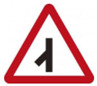 Forked road (9)	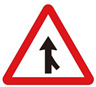 Entering roadway merge (right)
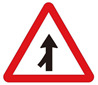 Entering roadway merge (left)	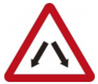 Divided lanes	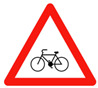 Careful, cyclists	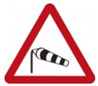 Careful, strong winds	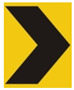 Safe orientation guidance

**Table 2 sensors-22-04768-t002:** Results of 23 classes.

Model	Precision	Recall	mAP
Mask R-CNN	78.40	29.57	28.64
YOLOv5	55.49	73.44	71.12
Our Approach	88.13	85.27	81.99

**Table 3 sensors-22-04768-t003:** Results of 5-fold cross-validation from Mask R-CNN.

Mask R-CNN	Precision	Recall	mAP
Fold 1	78.24	39.42	37.15
Fold 2	77.39	39.23	36.42
Fold 3	69.82	32.55	31.67
Fold 4	78.31	40.67	38.25
Fold 5	72.47	35.74	31.65
Average	75.25	37.52	35.03
Standard Deviation	3.47	2.98	2.81

**Table 4 sensors-22-04768-t004:** Results of 5-fold cross-validation from YOLOv5.

YOLOv5	Precision	Recall	mAP
Fold 1	56.22	74.17	72.81
Fold 2	55.40	73.92	72.04
Fold 3	54.82	73.36	71.94
Fold 4	55.27	74.03	72.14
Fold 5	54.98	73.45	72.48
Average	55.34	73.78	72.28
Standard Deviation	0.48	0.32	0.32

**Table 5 sensors-22-04768-t005:** Results of 5-fold cross-validation from Our Approach.

Our Approach	Precision	Recall	mAP
Fold 1	88.93	83.95	81.39
Fold 2	84.05	86.86	83.62
Fold 3	88.13	85.27	81.99
Fold 4	81.17	86.43	83.15
Fold 5	80.73	84.98	81.48
Average	84.60	85.50	82.33
Standard Deviation	3.41	1.04	0.90

**Table 6 sensors-22-04768-t006:** Results of training and testing images from same devices.

Model	Device	Precision	Recall	mAP
Mask R-CNN	Gopro	80.25	25.49	21.24
	Phone	75.72	34.69	32.70
YOLOv5	Gopro	54.49	77.45	71.19
	Phone	58.21	75.63	72.04
Our Approach	Gopro	71.49	66.86	61.05
	Phone	82.54	83.45	77.92

**Table 7 sensors-22-04768-t007:** Results of training and testing images from different devices.

Model	Train/Test	Precision	Recall	mAP
Mask R-CNN	Phone/Gopro	60.45	25.48	22.58
	Gopro/Phone	43.73	17.76	14.74
YOLOv5	Phone/Gopro	38.66	44.50	42.38
	Gopro/Phone	48.29	54.84	50.47
Our Approach	Phone/Gopro	79.94	62.32	58.34
	Gopro/Phone	76.81	79.11	72.13

**Table 8 sensors-22-04768-t008:** Results of all models in our approach.

Model	Precision	Recall	mAP
Mask R-CNN (Shape)	94.91	89.59	92.86
Xception (circle)	99.73	99.73	-
Xception (triangle)	98.45	98.30	-

**Table 9 sensors-22-04768-t009:** Results of different numbers of training images.

Model	Number of Training Images	Precision	Recall	mAP
Mask R-CNN	2000	77.31	39.89	35.94
	4000	81.37	46.22	43.32
	6000	77.30	38.65	35.25
	8000	86.47	45.27	43.18
	10,000	80.53	28.98	24.08
YOLOv5	2000	49.56	61.87	58.72
	4000	72.19	80.42	77.91
	6000	77.59	86.46	85.73
	8000	84.58	90.35	88.28
	10,000	92.24	96.94	94.63
Our Approach	2000	58.09	78.58	72.51
	4000	84.64	83.36	80.82
	6000	86.46	84.39	81.56
	8000	78.24	86.15	82.64
	10,000	92.53	89.73	88.72

**Table 10 sensors-22-04768-t010:** Results of varying image resolutions.

Mask R-CNN	Xception	Precision	Recall	mAP
1024 × 1024	224 × 224	88.13	85.27	81.99
512 × 512	112 × 112	66.06	57.00	47.17
1536 × 1536	336 × 336	76.50	91.12	84.34

**Table 11 sensors-22-04768-t011:** Results of varying image resolutions from YOLOv5.

Resolution	Precision	Recall	mAP
416 × 416	55.49	73.44	71.12
208 × 208	39.74	56.12	52.44
624 × 624	82.64	88.91	87.22

## Data Availability

The data presented in this study are available on request from the corresponding author. The data are not publicly available due to restriction of privacy.

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
