# Peer review of "A Hierarchical Approach for Traffic Sign Recognition Based on Shape Detection and Image Classification"

_sensors, 2022, doi:10.3390/s22134768_

Round 1

Reviewer 1 Report

The work requires the following corrections:

- first of all, it is worth adding a parameter, average accuracy, to the work,

- then make sure that the terms described in the paper are well named, e.g. in tab . 6, we have "Number of trainings" which refers to the number of training objects - and is not necessarily the number of trainings of the neural network.

- In order to compare with other methods, the training should be performed several times and the standard deviation should be shown, as it is, it is not entirely clear which method is the best.

- to say which method is best is only legitimate with appropriate statistical tests, as it stands it can only be written that the methods are comparable. 

- Finally, this work has potential, above all the authors spent a lot of time to form the training data set, which is definitely an added value. It would be worth extending the paper, describing in more detail the parameters used in the models and adding model codes to provide a reference point for other authors. As much as possible, after possible acceptance, make the used data available.

Author Response

Comments and Suggestions for Authors:

The work requires the following corrections:

  1. First of all, it is worth adding a parameter, average accuracy, to the work,

Response: Thanks for the good comments and suggestions. We have performed the 5-fold cross-validation and calculated the average precision, recall and mAP for all methods. All results have been added to page 12 of the paper, including Table 3, Table 4 and Table 5.

  1. Then make sure that the terms described in the paper are well named, e.g. in tab . 6, we have "Number of trainings" which refers to the number of training objects - and is not necessarily the number of trainings of the neural network.

Response: Thanks for the kindly reminder. We have corrected the description in Table 6 by changing "Number of trainings" to "Number of training images" (Table 9 in p. 13).

  1. In order to compare with other methods, the training should be performed several times and the standard deviation should be shown, as it is, it is not entirely clear which method is the best.

Response: Thanks for the good suggestions again. We have performed 5-fold cross-validation and calculate the standard deviation of precision, recall and mAP for all methods. The results show that the standard deviation of each fold is very low, it means that the performance of each method is very stable. All results have been added to page 12 of the paper, including Table 3, Table 4 and Table 5.

  1. To say which method is best is only legitimate with appropriate statistical tests, as it stands it can only be written that the methods are comparable.

Response: Thanks for the good comments. To verify that our experimental results are legitimate, we try to perform 5-fold cross-validation. However, each model takes at least 10 hours from training to testing. Since we only have 10 days to revise, we prioritize the main experiment, which is the first experiment. From the results, we can observe that there is no much difference in the results of each fold. It can indirectly show that other experimental results are reliable. All results have been added to page 12 of the paper, including Table 3, Table 4 and Table 5.

  1. Finally, this work has potential, above all the authors spent a lot of time to form the training data set, which is definitely an added value. It would be worth extending the paper, describing in more detail the parameters used in the models and adding model codes to provide a reference point for other authors. As much as possible, after possible acceptance, make the used data available.

Response: Thanks for the good suggestions. We have attempted to describe more details of the model parameters, which have been added on page 11 of this paper. Furthermore, the model code and dataset will be organized and prepared for future reference.

Reviewer 2 Report

The paper is well written, easy to read. The two-stage approach is a good choice. However, it has some flairs.

The number of traffic sign “classes” (23) in the survey seems to be quite low, considering the huge variety of the signs in the practice.

Many of the traffic signs contain text, but in the sample used, there are hardly text elements. Please give some comments on the possibilities of recognizing text information.

Another problem is the age structure of the literature survey. From the 26 sources only six are younger, then 5 years, 14 are between 6 and 10 years, while six are older than 10 years. Some of the references are popular webpages rather than scientific journals. One of the internet sources was accessed 5 years ago. In the field of image recognition more up-to date resources would be necessary. Due to the lack of up to date references it is difficult to judge the novelty of the content.

Author Response

Comments and Suggestions for Authors:

The paper is well written, easy to read. The two-stage approach is a good choice. However, it has some flairs.

  1. The number of traffic sign “classes” (23) in the survey seems to be quite low, considering the huge variety of the signs in the practice.

Response: Thanks very much for the recognition of our efforts and give us many valuable comments and suggestions. Indeed, the number of traffic sign classes in this article is slightly less, and we are considering to expand our dataset continuously in the future work (p. 15). To identify more traffic sign categories, we need to collect enough relevant images and annotate them, train the model after expanding the dataset, and finally adjust and optimize the model according to the results.

  1. Many of the traffic signs contain text, but in the sample used, there are hardly text elements. Please give some comments on the possibilities of recognizing text information.

Response: Thanks for the insightful comments. The contents of the Taiwan traffic signs are divided into geometric graphics, digits and Chinese characters. The difficulty ranges from simple to difficult: pure geometric graphics, pure digits, and Chinese characters. The most difficult to deal with is the composite sign, such as the sign with "where to go" seen on the highway, which will contain both graphics and text, which will be more difficult to deal with. In this paper, we first discuss the identification of traffic signs with geometric graphics. In the future, we will try to identify digits, Chinese characters, and even more difficult composite signs. Relevant descriptions have also been added to page 15 of this paper.

  1. Another problem is the age structure of the literature survey. From the 26 sources only six are younger, then 5 years, 14 are between 6 and 10 years, while six are older than 10 years. Some of the references are popular webpages rather than scientific journals. One of the internet sources was accessed 5 years ago. In the field of image recognition more up-to date resources would be necessary. Due to the lack of up to date references it is difficult to judge the novelty of the content.

Response: Thanks for the kindly reminders. Regarding the literature survey, we have supplemented the description of several articles in recent years in Section 2.3 (pp. 6-7). The 4 new articles use Mask R-CNN as the basic model are listed as follows. Additionally, all of the older articles refer either to the nuances of traffic safety regulations or to the progress in image recognition model development. The used methods have been introduced in recent years, one of them – YOLOv5 – does not even have the official paper released by the model’s authors. Current traffic sign detection/classification solutions are still based on traffic safety regulations introduced many years ago, these goals/concerns haven’t changed since then, and the challenges faced by modern traffic sign recognition solutions, introduced when the field of object detection and image classification was on the rise still remain to this day.

  • [25] Singh, et al. used Mask R-CNN for road damage detection, the authors achieved a mean F1 score of 0.528 with an IoU of 50% in real-world road images acquired with a smartphone camera.
  • [15] Malbog, et al. focused on pedestrians and used Mask R-CNN to detect pedestrian crosswalks.
  • [31] Xu, et al. presented a simple and effective Mask R-CNN algorithm for more rapid detection of vehicles and pedestrians. The authors adjusted the backbone network and FPN to effectively improve the speed and mAP.
  • [32] Zhang, et al. discussed on the topic of vehicular accidents, the author proposes a vehicle-damage-detection segmentation algorithm, which improves the efficiency of solving traffic accident compensation problems through self-made dedicated dataset and improved Mask R-CNN model.

Round 2

Reviewer 1 Report

The amendments made are satisfactory and I have no further comments.

Reviewer 2 Report

My concerns have been properly addressed.